# Liposomal Fluopsin C: Physicochemical Properties, Cytotoxicity, and Antibacterial Activity In Vitro and over In Vivo MDR *Klebsiella pneumoniae* Bacteremia Model

**DOI:** 10.3390/antibiotics14090948

**Published:** 2025-09-19

**Authors:** Mickely Liuti Dealis Gomes, Leandro Afonso, Kawany Roque Basso, Leonardo Cruz Alves, Enri Josué Navia Macías, Sueli Fumie Yamada-Ogatta, Ana Carolina Guidi, João Carlos Palazzo de Mello, Fábio Goulart Andrade, Luís Fernando Cabeça, Martha Viviana Torres Cely, Galdino Andrade

**Affiliations:** 1Microbial Ecology Laboratory, Department of Microbiology, State University of Londrina, Londrina 86057-970, Brazillafonso.bio@gmail.com (L.A.); kawroquebasso.555@uel.br (K.R.B.); leonardo.cruz.alves@uel.br (L.C.A.); enrim238@gmail.com (E.J.N.M.); andradeg@uel.br (G.A.); 2Laboratory of Molecular Biology of Microorganisms, Department of Microbiology, State University of Londrina, Londrina 86057-970, Brazil; ogatta@uel.br; 3Laboratory of Animal Experimentation, Department of Pharmacy, State University of Maringá, Maringá 87020-900, Brazil; acguidi2@gmail.com (A.C.G.); mello@uem.br (J.C.P.d.M.); 4Laboratory of Histopathological Analysis, Department of Histology, State University of Londrina, Londrina 86057-970, Brazil; andrade@uel.br; 5Laboratory of Biomaterials and Organic Biomolecules, Department of Chemistry, Federal Technological University of Paraná, Londrina 86036-700, Brazil; luiscabeca@utfpr.edu.br; 6Agricultural and Environmental Sciences Institute, Mato Grosso Federal University, Sinop 78550-728, Brazil

**Keywords:** natural product, antimicrobial, toxicity, liposomes, *Pseudomonas aeruginosa* LV, encapsulation

## Abstract

**Introduction:** Antimicrobial resistance has become a global concern, and few new antimicrobials are currently being developed. Fluopsin C has proven broad-spectrum activity, being a promising candidate for new antimicrobial development. To optimize antimicrobial activity, this research aimed at fluopsin C (Flp) encapsulation in liposomes to achieve controlled release and reduce cytotoxicity. **Methods:** Liposomal formulations were prepared by extruding formulations based on soy phosphatidylcholine (SPC) or poly (ethylene glycol)-distearoylphosphatidylethanolamine (DSPE-PEG) plus cholesterol, and were characterized by their size, polydispersity index, zeta potential, encapsulation efficiency, shelf-life stability, in vitro release profile, cytotoxicity, and antimicrobial activity against *Klebsiella pneumoniae* in vitro and in vivo. **Results:** The results indicated that the DSPE-PEG DMSO+Flp formulation presented superior physicochemical stability and unaltered antimicrobial activity. In vitro, CC_50_ decreased by 54%. No lethal dose was obtained in mice within the concentration range tested. The most effective doses in vivo were 2 × 2 mg/kg for free fluopsin C and 1 × 2 mg/kg for DSPE-PEG DMSO+Flp, resulting in a 40% reduction in mortality from bacteremia. Only discrete inflammatory infiltration was detected in the liver, while kidney necrosis ranged from discrete to moderate. Encapsulation of fluopsin C in liposomes showed promising features supporting to use against infections by MDR *K. pneumoniae*.

## 1. Introduction

Among infections caused by multidrug-resistant (MDR) microorganisms, the World Health Organization (WHO) classified carbapenem-resistant Enterobacterales (CRE), including *Klebsiella pneumoniae*, as critical in terms of the urgency for developing new antibiotics [1,2]. *K. pneumoniae* is particularly associated with severe bloodstream infections and secondary pneumonia in patients with viral respiratory infections [3].

The search for new MDR-suppressing antimicrobials includes overcoming current approaches such as combinatory therapies and dosage augmentation, usage reformulations of already commercialized drugs, and developing new drugs with novel modes of action [4,5,6,7,8]. Fluopsin C corresponds chemically to bis(N-methylthioformohydrox-amato) Cu(II)) and is a natural broad-spectrum antimicrobial discovered five decades ago. It is a low-molecular-weight compound produced by the secondary metabolism of *Pseudomonas* spp. and *Streptomyces* sp. grown in the presence of copper [9]. This compound can be purified from microbial culture supernatants as thin dark-green-brownish prismatic crystals and shows antibiotic activity against MDR strains, in vitro and in vivo [9,10,11,12]. Kerbauy et al. [11] tested FlpC against 69 MDR-*K. pneumoniae* isolates, both KPC producers and resistant to colistin and polymyxin B, including under biofilm conditions, resulting in MICs of 1.95 to 3.9 µg/mL.

In addition, a low frequency of Fluopsin C-resistant mutants was observed, as well as an efficient antibiofilm activity [9,10,11]. However, the application of the compound was discouraged due to a high toxicity found in in vitro and in vivo experiments, still considered unviable for clinical purposes [10,11,12,13,14], in addition to the poor pharmacokinetic properties [10].

Several antimicrobials exhibit low or moderate cytotoxicity and remain essential resources to control MDR infections [14]. In this context, we believe that developing formulations for fluopsin C may be a key to enhancing its potential as a new drug development. Liposomal encapsulation is a promising strategy to reduce the compound toxicity. This system-controlled drug release is widely explored in pharmacology, improving treatments for high-risk pathogens while minimizing adverse effects found in conventional formulations [15,16]. This method shows some advantages, include enhancing the therapeutic effects, controlling drug release, reducing toxicity, enhancing pharmacokinetics and solubility, and reducing the doses required to maintain the therapeutic effect [15,17,18,19].

The objectives of this paper were to evaluate the effect of liposomes encapsulation of fluopsin C on the cytotoxicity and the pharmacokinetic properties of this compound.

## 2. Results

### 2.1. Characterization of the Fluopsin C-Containing Liposomal Formulations

Four liposomal formulations were developed for comparison of their physicochemical features. All formulations followed the ratio of total lipids: fluopsin C (Flp) at 1:1 Mm and are based on soy phosphatidylcholine (SPC), distearoyl phosphatidyl ethanolamine-polyethyleneglycol (DSPE-PEG), and cholesterol in different concentrations that have been proven over time.

Data obtained from the initial physical–chemical characterization of liposomes containing fluopsin C (Figure 1) showed that at time zero, all formulations presented a polydispersity index (PDI) below 0.3. Regarding the liposomal hydrodynamic diameter, DSPE-PEG-containing formulations were smaller than those containing only SPC. Membrane charges measured by zeta potential (ZP) were higher in formulations with dimethylsulfoxide (DMSO) than in those without this fluopsin C solvent, specifically SPC DMSO+Flp and DSPE-PEG DMSO+Flp, which were negatively charged.

The stability of the formulations was assessed at specific time zero (up to 30 days after production), 3, 6, and 12 months after production. DSPE-PEG formulations presented smaller diameters in the long term, especially when stored in solutions. Overall, ZP charges tended to decrease over time; after 12 months, all formulations showed ZP values above −10 mV. Regarding the PDI, all formulations in all storage conditions did not exceed the value of 0.3, considered a good stability parameter for large-scale production.

The fluopsin C-containing liposomal formulations and their respective liposomal controls were analyzed through scanning electron microscopy (SEM) at time zero, revealing a high degree of deformity in the control formulations (i.e., without incorporation of fluopsin C). In contrast, fluopsin C-containing liposomes displayed a spherical shape with morphological uniformity, consistent with the results determined by dynamic light scattering (DLS).

The incorporation of fluopsin C in formulations DMSO-containing enhanced the best encapsulation efficiency (Table 1), around 80%.

Fluopsin C release kinetics in vitro were monitored for the free compound, and the DMSO-containing formulations (SPC DMSO+Flp and DSPE-PEG DMSO+Flp) (Figure 2). After 11 h, 87% of free fluopsin C was released. In the liposomal formulations, at the same time, the value reached 63% for SPC DMSO+Flp and 52% for DSPE-PEG DMSO+Flp of the available 1 mM. Considering the EE% in the release of liposomal fluopsin C, there was a gradual release of encapsulated material over the first 11 h, 11.88 ± 1.69 µg/h for SPC DMSO+Flp, 9.81 ± 1.88 µg/h for DSPE-PEG DMSO+Flp compared to 19.32 ± 2.44 µg/h for free fluopsin C. After the first 11 h, the release of fluopsin C is minimal, leading to a maximum release in 24 h of 67% for SPC DMSO+Flp, 55% for DSPE-PEG DMSO+Flp and 91% for free fluopsin C, considering the limitations of the release technique by Franz cells.

### 2.2. Determination of In Vitro Liposomal Fluopsin C Cytotoxicity

Cytotoxicity in LLC-MK2 cells was assessed for free fluopsin C, the liposomal formulations SPC DMSO+Flp and DSPE-PEG DMSO+Flp, and their respective control (Table 2). Compared to the free fluopsin C, DSPE-PEG DMSO+Flp reduced the cytotoxicity 54% and 48% in the CC_50_ and CC_90_ values, respectively. Meanwhile, SPC DMSO+Flp reduction observed was 69% and 88% in CC_50_ and CC_90_, respectively. The DSPE-PEG DMSO liposomal control did not affect cell viability, even at the highest concentration tested, with 100% of the cells viable (Appendix A Figure A1).

### 2.3. In Vivo Assessment of Liposomal Fluopsin C Toxicity

No mortality of Swiss mice was observed in all the concentrations tested; therefore, the lethal dose of DSPE-PEG DMSO+Flp was estimated as >8 mg/kg. In a previous study conducted by our research group, under similar experimental conditions [10], the lethal dose (LD_50_) of free fluopsin C was estimated at 4 mg/kg. These data indicate that the liposomal formulation reduced the toxicity of the compound in mice. Histopathological analyses were performed for assessment of acute toxicity (single dose, *n* = 12) and repeated dose toxicity (3 doses every 8 h, *n* = 12), in healthy animals using the most effective dose of 1 × 2 mg/kg DSPE-PEG DMSO+Flp intravenous [20]. Histopathological analysis of the liver revealed no hemorrhage at any time analyzed. However, mild to moderate necrosis and inflammatory infiltrate were present in all groups, including negative and liposomal controls (Appendix A Table A1). The only significant difference in acute toxicity class occurred for hepatocytes’ nuclei area (*p* = 0.04), though similar values were found in the control group. Treatments did not alter single-nucleus hepatocyte number. In terms of repeated dose toxicity, there are significant differences in 40 days in the number of double-nuclei hepatocytes, increasing along the tested time. In the same way, this fact occurred over time and was found in all treatments, except the negative control, which maintained similar means throughout the study, suggesting that the liver may recover following the antimicrobial treatment (Figure 3).

Nephrotoxicity analysis revealed the presence of mild to moderate necrosis, without inflammatory infiltration or hemorrhage in all treatments (Appendix A Table A1). Significant differences were found when comparing Bowman’s space area on day one between the acute toxicity (*p* = 0.002) and repeated doses toxicity groups (*p* = 0.003), with the negative control. This indicates that, on the administration day, the Bowman’s space area in the treated groups was contracted, while in the control a space area was found two time bigger. After time zero, no significative differences were observed, indicating the recovery of the organ to normal metrics (Figure 4). Additionally, proximal and distal tubule diameters remained unaltered, with consistent peripheral nucleus placement across all groups and time points.

### 2.4. Determination of the Minimum Inhibitory Concentration (MIC) and Minimum Bactericidal Concentration (MBC) of Free and Liposomal Fluopsin C

Fluopsin C showed antibacterial activity against the *K. pneumoniae* strains tested. MIC and MBC values showed no significative difference for both strains, when compared to fluopsin C and fluopsin C combined with liposome. For *K. pneumoniae* ATCC 10031 [10], the sensitive strain, the MIC and MBC values were both equal to 1.66 µg/mL, while for *K. pneumoniae* KPN-19, the MDR strain, they were both 3.32 µg/mL. No variance was found in the MIC/MBC tests, performed in triplicate.

### 2.5. Assessment of the Effect of Free and Liposomal Fluopsin C over the Bacteremia Model in Mice

Lethal bacterial inoculum for 100% of the animals using *K. pneumoniae* KPN-19 was achieved with 10^8^ UFC/mL, 20 h after infection. Bacteremia was detected from the first blood collection (25 CFU/50 µL blood). Post-mortem organs (lungs, kidneys, and liver) assessment also indicated the generalized presence of the bacteria. This inoculum cell density was standardized for all further experiments.

The dose of 2 × 2 mg/kg of free fluopsin C (Figure 5) led to a 20% mortality decrease after 24 h, compared to the other doses. The DSPE-PEG DMSO+Flp formulation provided the best outcome at 1 × 2 mg/kg, reducing mortality by 40% in the bacteremia in mice. Both were considered the most effective doses for free and encapsulated forms.

## 3. Discussion

Liposomal encapsulation to optimize antimicrobial performance is one of the most promising approaches currently available [15], being particularly beneficial to avoid antimicrobial resistance [21,22]. Fluopsin C showed promise for broad-spectrum antimicrobial activity; however, the pharmacokinetics are low understanding, and high toxicity is a challenge during drug development, still considered unviable for clinical purposes [9,10,11,12,13,14]. Liposomal encapsulation should minimize the high cytotoxicity effects observed of fluopsin C.

In the present study, the DSPE-PEG DMSO+Flp formulation showed the best physical-chemical properties from production to 12 months of storage, especially when stored in solution at low temperature. The formulation preserved the characteristics over time, with predicted variations in size and ZP, but not surpassing the desirable limits to use as an antimicrobial (<500 nm, >±10 mV, and ≤0.3 of PDI) [23]. A PDI value below 0.3 indicates monodispersity, and the homogeneity of size was confirmed using SEM [24].

The decrease in PZ represents a decrease in electrostatic repulsion, with an increased tendency to aggregate, potentially leading to an increase in hydrodynamic diameter. Therefore, the DSPE-PEG DMSO+Flp formulation, both lyophilized and stored in solution, maintained greater PZ stability and, thus, a less variable size. For the SPC DMSO+Flp formulations (in solution and lyophilized), there was a pronounced increase in PZ, resulting in greater agglomeration and a proportional increase in particle diameter [25].

After 11 h, the release of liposomal fluopsin C (DSPE-PEG DMSO+Flp) reached 52%. Similar results were reported, and higher cholesterol content resulted in slower drug release over time, extending the duration of vancomycin and rifampicin release [26]. Cholesterol enhances vesicle stability and reduces membrane permeability, helping minimize cold-induced damage during lyophilization [27,28]. These results are in accordance with the findings of Scriboni et al. [29] that reported values of 30–70% of vancomycin release in 10 h, depending on the formulation tested, which was a similar method used in this paper.

In the in vivo system, a higher difference in release was expected among liposomal formulations, due to the presence of plasmatic proteins, increasing the potential of the DSPE-PEG DMSO+Flp formulation. In the absence of PEG of liposomal formulation, plasmatic proteins opsonize the liposomes, leading to clearance by the mononuclear phagocytic system [28,30]. Previous studies on antimicrobial liposome encapsulation have found similar results to ours, including encapsulation efficiency [31,32,33,34].

The SPC DMSO+Flp formulation with a high concentration of fluopsin C, presented CC_50_ and CC_90_ values 69% and 88% higher than free fluopsin C. However, according to other authors, this formulation could lead to fast opsonization, phagocytosis, and removal from circulation by the action of the mononuclear phagocyte system, presenting a potential risk for this formulation [30]. DSPE-PEG DMSO+Flp was identified as the most promising formulation due to the advantages over the immune system, besides achieving CC_50_ and CC_90_ values 54% and 48% higher than free fluopsin C, respectively. These advantages were also observed by other Authors [17]. In vivo, there was no mortality associated with the use of this formulation at 8 mg/kg, whereas free fluopsin C showed a DL_50_ of 4 mg/kg in mice [10].

Liver toxicity was detected as a mild inflammatory infiltrate, which was also found by Navarro et al. [10]. However, complete recovery of the organ was observed 40 days after administration of liposomal fluopsin C (Appendix A Table A1). The nephrotoxicity profile differed from the one found for free fluopsin C [10], with the presence of necrosis (including in the liposomal control), but without other types of lesions.

No alterations in antimicrobial activity were detected between free and liposomal fluopsin C in vitro, indicating that the encapsulation technique did not suppress this activity. Our findings are similar to Kerbauy et al. [11], with 1.95 µg/mL for ATCC 10031 and Navarro et al. [10], with 1 µg/mL for ATCC 10031 and 2 µg/mL for KPN-19. In previous studies by our group, treatment of mice infected with MDR *K. pneumoniae* using 2 mg/kg of free fluopsin C improved survival by 20% at 96 h [10]. Here, under similar infection conditions, we obtained a 40% reduction in mortality for the same dose at 24 h.

## 4. Materials and Methods

### 4.1. Microorganisms

In this study, the target microorganism used was the human pathogen *K. pneumoniae* KPN-19, isolated from tracheal secretion, and *K. pneumoniae* ATCC 10031 strain as a control. Both strains were provided by the Laboratory of Basic and Applied Bacteriology of the State University of Londrina, which were used in a previous study [10,11]. KPN-19 strain carries the *bla*_KPC_ gene, presents porin loss, and resistance to many antibiotics such as imipenem, meropenem, ertapenem, polymyxin B, and colistin [11]. Fluopsin C-producing *Pseudomonas aeruginosa* LV strain (GenBank CP058323.1) was isolated from citrus canker lesions of orange leaves in an orchard at Astorga, Paraná, Brazil [35,36]. All strains were cryopreserved in 40% (*v*/*v*) glycerol solution in liquid nitrogen and deposited in the Laboratory of Microbial Ecology bank of microorganisms.

### 4.2. Production and Purification of Fluopsin C

*P. aeruginosa* LV was activated by two consecutive cultures of 48 and 24 h in a nutrient agar plus 100 mg/L of CuCl_2_·2H_2_O, and incubated at 28 °C. The fluopsin C production process was performed according to the patented procedure PI 0803350-1 [37], with the modifications proposed by Afonso et al. and Bedoya et al. [9,38]. Purification of fluopsin C was conducted using chromatographic processes, as described by Navarro et al. [10]. High-performance liquid chromatography analysis employed an Agilent 1260 Infinity (Agilent Technologies, Santa Clara, CA, USA) equipped with a multi-wavelength detector and a Zorbax SB-C18 column (4.6 × 250 mm, 5 µm) (Agilent Technologies, Santa Clara, CA, USA), as described by Bedoya et al. [38]. The purity level of fluopsin C used was higher than 80% (Appendix A Figure A2).

### 4.3. Production and Extrusion of Lipid Vesicles

Four liposomal formulations were used to compare the physicochemical features (Table 3). All formulations followed the amount of total lipids as follows: fluopsin C (Flp) 1:1 mM and are based on soy phosphatidylcholine (SPC), poly(ethylene glycol)-distearoylphosphatidylethanolamine (DSPE-PEG), and cholesterol in different concentrations. Liposomal formulations without fluopsin C were used as control (i.e., liposomal control) (named DSPE-PEG DMSO, DSPE-PEG, SPC DMSO, and SPC).

For each formulation, the respective lipids were aliquoted from chloroform stock solutions according to the concentrations described in Table 3. The samples were left at room temperature for chloroform evaporation, forming a lipid film. DSPE-PEG DMSO+Flp and SPC DMSO+Flp formulations, plus fluopsin C initially, were diluted in DMSO 10% and were added to phosphate buffer (0.1 M, pH 7.4), suspending the emulsion and promoting the incorporation of fluopsin C. The other formulations (DSPE-PEG+Flp and SPC+Flp) had active incorporation of fluopsin C, added during the formation process of the lipid film. After that, phosphate buffer was added to generate liposomal vesicles without DMSO addition.

After resuspended, the formulations were homogenized using a vortex (XH-D Vortex, Global Trade Technology, Jaboticabal, Brazil) for 1 min (min) and sonicated (Ultrasonic Washer L200, Schuster, Santa Maria, Brazil) for 1 min, forming large multilamellar liposomal vesicles. To obtain smaller liposomes, extrusion was performed using a mini extruder (Avanti Polar Lipids, Inc., Alabaster, AL, USA) with polycarbonate membranes (0.4 µm, 19 mm) (Avanti Polar Lipids, Inc., Alabaster, AL, USA), for 15 cycles at room temperature.

### 4.4. Characterization of Fluopsin C-Containing Liposomes

Fluopsin C-containing liposome formulations were characterized by determination of zeta potential (ZP), polydispersity index (PDI), hydrodynamic diameter, encapsulation efficiency, and fluopsin C release curve. For all tests, 1 mM liposomal fluopsin C was used. The dynamic light scattering (DLS) was used to determine the mean liposome diameter, PDI, and ZP using a NanoPlus particle analyzer (Micromeritics Instrument Corporation, Norcross, GA, USA). PDI below 0.3 was considered monodisperse [24]. Analysis of ZP, PDI, and liposomal diameter was performed four times: 0, 3, 6, and 12 months of storage at 10 °C. Liposomes were stored in solution and lyophilized. All measurements were performed in triplicate.

#### 4.4.1. Encapsulation Efficiency

To determine encapsulation efficiency, the liposomal formulations were centrifuged in a Minispin (Eppendorf F45-12-11 rotor, Merck, Darmstadt, Germany) at 12,100× *g* (G-Force) for 1 h. The Amicon Ultra-0.5 mL centrifugal filter unit (Ultracel-10 regenerated cellulose membrane, 10,000 NMWL, Merck, Darmstadt, Germany) was used for liposome retention. The measurement of free fluopsin C in solution (non-encapsulated) was performed by spectrophotometry [39] (BioMate3, Thermo, Waltham, MA, USA) at λ = 264 nm, considering the standard method of quantification [40]. The absorbance was compared with a fluopsin C calibration curve, for formulations in the presence or absence of DMSO (range of 13 a 1 μg/mL, r^2^ = 0.99). The encapsulation efficiency (EE%) was determined by Equation (1) [26], and total fluopsin C was set as 1 mM, and free fluopsin C in the solution was the amount retained in solution after liposomes removal by ultracentrifugation.(1)$$EE\%=\left(\frac{\left[Total\;Fluopsin\;C-free\;Fluopsin\;C\right]}{Total\;Fluopsin\;C}\right)\times 100$$

#### 4.4.2. Determination of Free and Liposomal Fluopsin C Release Curve

Free and liposomal fluopsin C released were determined to select the best formulation. Franz cells filled with phosphate plus 10% DMSO *v*/*v* in constant agitation were maintained at 37 °C, ensuring sink condition [28]. Fluopsin C release was analyzed by spectrophotometry (BioMate3, Thermo, Waltham, MA, USA) at λ = 264 nm. Aliquots of 300 μL were taken from the Franz cells at 15 and 30 min, 1, 2, 4, 6, 12, and 24 h. The absorbance was used to determine fluopsin C concentration. The experiments were performed in triplicate.

### 4.5. Analysis of Liposomal Formulations by Scanning Electron Microscopy (SEM)

Samples of lyophilized liposomal formulations were fixed in glutaraldehyde 2.5% fixing solution for 30 min. After that, samples were washed three times with sodium cacodylate buffer 0.1 M, pH 7.2, supplemented with paraformaldehyde solution 2%.

After washing, osmium tetroxide 1% was added, and the samples were incubated for 30 min in the dark to preserve lipid conservation. After that, the samples were washed as described above. For dehydration, samples were treated with serial immersions in different concentrations of ethanol (70%, 90%, and 100%) for 10 min. The last concentration was repeated twice. After that, samples were dried at the critical point using CO_2_ on a Critical Point Dryer 030 (BalTec Ag, Pfäffikon, Switzerland), coated with gold using the SDC 050 Sputter Coater (BalTec Ag, Switzerland), and observed at FEI quanta 200 scanning electron microscope (FEI Company/Thermo Fisher Scientific, Hillsboro, OR, USA) operating at 30 kV.

### 4.6. Determination of In Vitro Liposomal Fluopsin C Cytotoxicity

Likewise performed by Navarro et al. [10] and Kerbauy et al. [11], the LLC-MK2 cell line (renal epithelial cells of *Macaca mulatta*, CCL-7, ATCC, 20–25 passage, Merck Co, São Paulo, Brazil) was grown in a 96-well plate in RPMI medium supplemented with fetal bovine serum 10%, with 2.5 × 10^4^ cells/well and incubated for 24 h at 37.0 °C CO_2_ 5.0% [10,11,41]. A 100% confluence, non-adherent cells in suspension were removed by washing with sterile phosphate-buffered saline (PBS, 0.1 M, pH 7.2).

Formulation DSPE-PEG DMSO+Flp, SPC DMSO+Flp, and the respective control were serially diluted in RPMI medium, at concentrations ranging from 32 to 0.25 μg/mL of fluopsin C, and subsequently added to the cell-containing wells. The treatments were incubated for 24 h under the same conditions mentioned earlier. Cell viability was determined by the MTT test (3-[4,5-dimethylthiazol-2-yl]-2,5-diphenyl-2H-tetrazolium bromide) (Sigma Chemical Co., St. Louis, MO, EUA) reduction method, according to the manufacturer’s recommendation. A non-linear regression method was performed to determine the cytotoxic concentration that inhibits 50% (CC_50_) and 90% (CC_90_) of the metabolic activity of the cell population in 24 h.

### 4.7. Determination of the Minimum Inhibitory and Bactericidal Concentration of Free and Liposomal Fluopsin C

Minimum inhibitory concentration (MIC) and minimum bactericidal concentration (MBC) were determined for DSPE-PEG DMSO+Flp and SPC DMSO+Flp formulations against *K. pneumoniae* ATCC 10031 and KPN-19 strains. Tested concentrations ranged from 13.3 to 0.2 µg/mL, including for free fluopsin C. Controls consisted of the respective formulations without the incorporation of fluopsin C. The MIC values were determined through the broth microdilution method following the Clinical and Laboratory Standards Institute (CLSI) guidelines of 2021 [42], and incubated for 24 h at 37 °C. For MBC determination, 10 µL aliquots from each well of the microdilution method were plated onto Mueller-Hinton Agar and incubated for 24 h at 37 °C.

### 4.8. In Vivo Assessments

The in vivo experimental design was carried out as shown in Figure 6.

#### 4.8.1. Animals

The following experiments utilized 42-day-old female Swiss mice, with an average weight of 25 g (total *n* = 189). These animals were provided by the Central Bioterium of the Biological Sciences Center at the State University of Londrina and the Central Bioterium of the State University of Maringá. Ethical approval for animal use was granted by the respective ethical commissions of the respective universities under the registrations CEUA (Comitê de Ética no Uso de Animais) n° 039.2021 and CEUA n° 9838190121, respectively.

Compliance with regulations concerning animal welfare and management was ensured throughout the study. The animals were randomly allocated to the different experimental groups using a simple randomization protocol. The researchers responsible for administering the treatments/inoculum and assessing the clinical outcomes (including signs of infection and mortality) were blinded as to the identity of the groups. After being assigned to the groups, the animals were kept for a period of 3 days, for adaptation in the new environment, before each experiment. To minimize confounding factors, the cages were kept in ascending order according to the corresponding dose, and the control groups were segregated on another shelf, with greater spacing from the others, avoiding cross-contamination. For the same purpose, the animals were identified with a number according to the cage number, in a painless way.

#### 4.8.2. Lethal Inoculum of *K. pneumoniae* KPN-19 and Bacteremia Model Establishment

For the determination of the lethal inoculum, *n* = 5 animals were infected intraperitoneally with *K. pneumoniae* KPN-19 (0.5 mL) at different cell densities of 10^7^, 10^8^, and 10^9^ CFU/mL. The control group (*n* = 5) was administered with sterilized saline solution. The animals were observed until 96 h post-inoculation, for a total of *n* = 20 animals.

After the previous results, the bacteremia model was confirmed by infecting *n* = 5 animals with 10^8^ CFU/mL, intraperitoneally. 50 µL of caudal blood was collected post-anesthesia (isoflurane 3–5% mg/kg by inhalation) every 2 h after inoculation. Time zero was defined as 10 min after inoculation. The blood samples were plated on blood agar and incubated at 37.0 °C for 24 h. After euthanasia, the lungs, kidneys, and liver were collected. The organs were weighed and macerated, and the product was serially diluted in sterilized PBS. Aliquots were then plated on blood agar and incubated at 37.0 °C for 24 h. The same procedures were performed for control group animals (*n* = 5). for a total of *n* = 10 animals.

#### 4.8.3. Lethal Dose for Fluopsin C Liposomal Formulation

In a previous study, we evaluated the lethal dose of free fluopsin C in female Swiss mice [10]. To compare the effect of the liposomal formulation, we used similar experimental conditions. Therefore, the animals were inoculated intravenously with single doses of 0.5, 1, 2, 4, and 8 mg/kg DSPE-PEG DMSO+Flp formulation. The control group received sterilized PBS. The experimental groups consisted of *n* = 6 animals kept in individual containers, for a total of *n* = 36 animals. The animals were monitored until 96 h post-administration.

#### 4.8.4. Antibacterial Activity of Liposomal and Free Fluopsin C In Vivo

To determine the protective effects of liposomal and free fluopsin C, *n* = 5 animals were infected with 0.5 mL 10^8^ CFU/mL of *K. pneumoniae* KPN-19 intraperitoneally, according to Navarro et al. [10]. Immediately after inoculation, 0.1 mL of free fluopsin C or DSPE-PEG DSMO+Flp formulation was administered intravenously. Different dosages of both compounds were evaluated as single doses of 1 × 0.5, 1 × 1, and 1 × 2 mg/kg and double doses with an 8 h interval of 2 × 0.5, 2 × 1, and 2 × 2 mg/kg, totaling 6 groups for each one. DSPE-PEG DMSO control formulation was used as the liposomal control (*n* = 5) at 2 × 2 mg/kg dose since there was no antimicrobial activity observed in vitro for the liposomal components. The negative control group (*n* = 5) consisted of the administration of sterilized saline solutions intraperitoneally and PBS intravenously. The positive control group (*n* = 5) consisted of the administration of sterilized PBS intravenously, for a total of *n* = 75 animals.

#### 4.8.5. Histopathological Analysis of the Most Effective Dose

Histopathological analyses were performed for assessment of acute toxicity (single dose, *n* = 12) and repeated dose toxicity (3 doses every 8 h, *n* = 12) [20]. Thus, healthy animals were administered DSPE-PEG DMSO+Flp intravenous using the most effective dose (1 × 2 mg/kg). Controls were based on administration of the same liposomal formulation without the addition of fluopsin C (*n* = 12) and sterilized PBS (*n* = 12). for a total of *n* = 48 animals.

The effects of the compounds administered were assessed 1-, 10-, 20-, and 40 days post-treatment. At each point, 3 animals were euthanized with the collection of the liver and kidney, which were immediately fixed using a 4% paraformaldehyde solution. The organs were processed to the obtention of histological slides and stained with hematoxylin/eosin.

Analysis of the organ’s sections was performed in agreement with Navarro et al. [10], consisting of the determination of hepatocyte nucleus area (*n* = 100); mono- and binucleated number of hepatocytes, number of hepatocytes with condensed chromatin and vacuolization [10]. Nephrotoxic effects were evaluated by determining the Bowman’s space area (*n* = 10), distal and proximal tubule diameters (*n* = 50), and nucleus placement of tubules lining cells. The presence of inflammatory infiltrates, hemorrhage, and necrosis was also determined for both organs. This qualitative evaluation of the histological results was carried out by a trained committee unrelated to the experiments, with samples identified by code.

### 4.9. Statistical Analysis

The results were averaged and are presented with the standard deviation. Liposomal cytotoxicity in vitro assays were analyzed through one-way ANOVA and Dunnett’s multiple comparison test; non-linear regression was also used to infer the CC_50_ and CC_90_. The Kaplan–Meier estimator was applied to generate a survival curve. Histological section images were processed using ImageJ 1.54d (Wayne Rasband and contributors National Institutes of Health, Bethesda, MD, USA) and analyzed through two-way ANOVA and Tukey’s multiple comparison test. All tests considered *p* < 0.05 as indicative of significative differences. Raw data were processed using GraphPad Prism v. 8.4.3 (GraphPad Software, Boston, MA, USA). The study did not have humane endpoints.

All animals were included throughout the experiment, and no experimental units or data points were excluded from the analysis. The experiment was conducted, the results were evaluated, and the data were analyzed by the same team, except for the qualitative evaluation of the histopathological analysis, where an independent analysis-trained committee unrelated to the experiments was formed for this purpose only, with samples identified by code.

## 5. Conclusions

The present study reported for the first time the effect of encapsulation of fluopsin C, using the liposomal technique. Liposomal fluopsin C (DSPE-PEG DMSO+Flp) was effective in controlling MDR *K. pneumoniae*, in vitro and in vivo in mice, increasing survival of infected animals. In addition, the reduction of cytotoxicity in vitro and in vivo conditions showed healthy mice and an increase in the in vitro release time of fluopsin C encapsulated. Based on experimental data and previous studies by our group, fluopsin C should be a promising compound for developing a new drug due to its broad activity against MDR. However, further study should be carried out in another animal model, according to preclinical study guidelines, to understand how the antimicrobial activity of fluopsin C does against MDR *K. pneumoniae*.

## Figures and Tables

**Figure 1 antibiotics-14-00948-f001:**
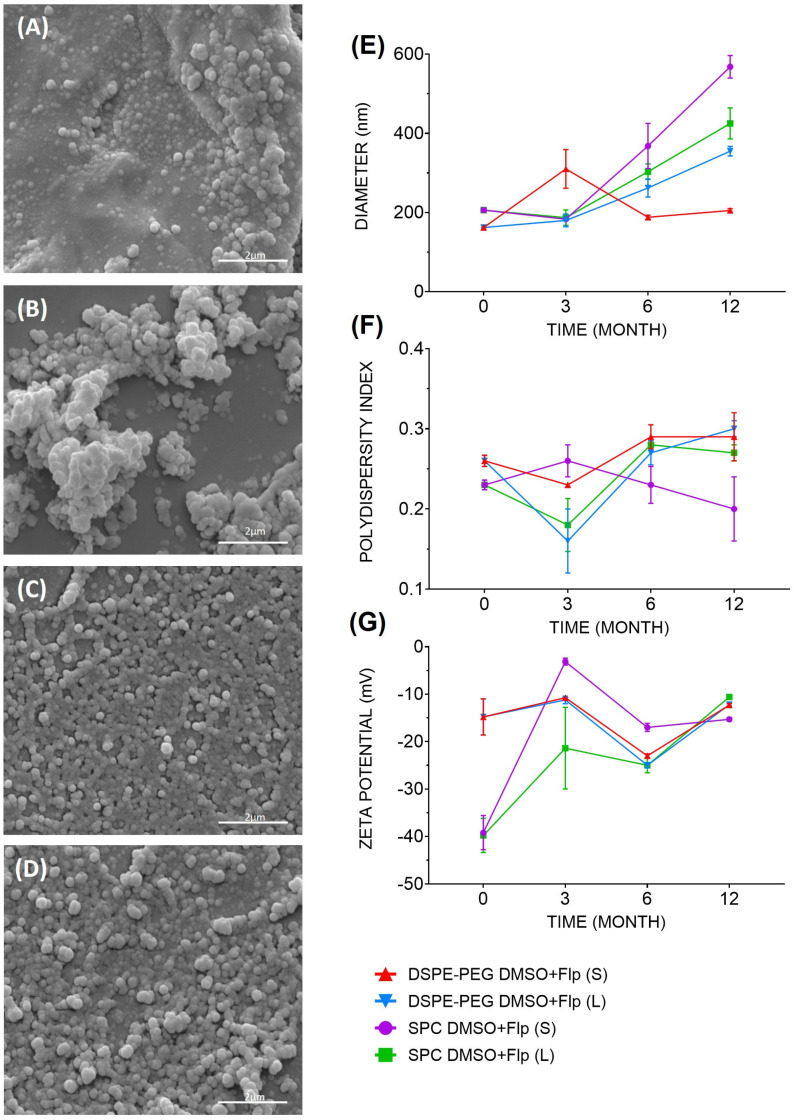
Formulation morphology and their respective controls were analyzed by scanning electron microscopy (SEM) under 30,000× magnification at time zero: (**A**) DSPE-PEG DMSO, (**B**) SPC DMSO, (**C**) DSPE-PEG DMSO+Flp, and (**D**) SPC DMSO+Flp. Physical–chemical characterization of fluopsin C-containing liposomal formulations. (**E**) Hydrodynamic diameter (nm), (**F**) polydispersity index (PDI), and (**G**) zeta potential (ZP, mV) were obtained by NanoPlus particle analyzer right after production and after 0, 3, 6, and 12 months. Formulations were stored in the form of solution (S) and lyophilized (L).

**Figure 2 antibiotics-14-00948-f002:**
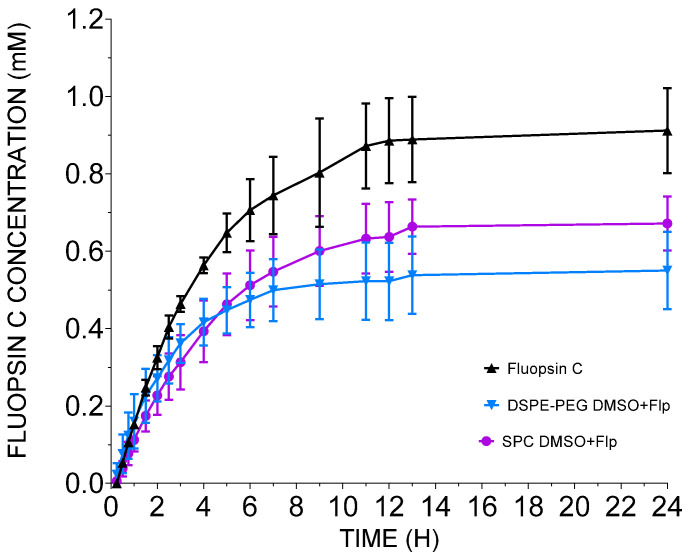
In vitro release curve of free and liposomal fluopsin C formulations using Franz cells, over 24 h.

**Figure 3 antibiotics-14-00948-f003:**
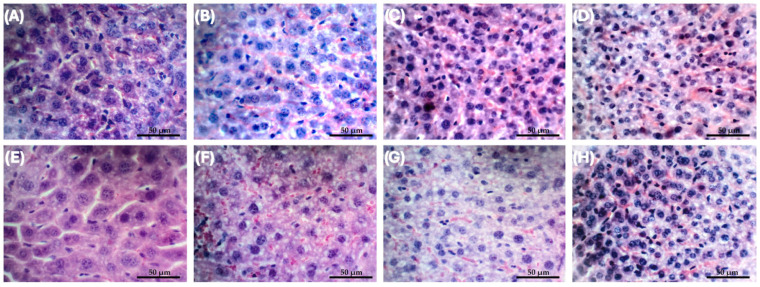
Liver analysis of repeated dose toxicity (RDT) and negative control (NC) over 40 days post-treatment, in the number of double-nuclei hepatocytes. (**A**) NC in day one, (**B**) NC in 10 days, (**C**) NC in 20 days, (**D**) NC in 40 days, (**E**) RDT in day one, (**F**) RDT in 10 days, (**G**) RDT in 20 days, and (**H**) RDT in 40 days. Color: hematoxylin/eosin (HE). Scale bar: 50 µm.

**Figure 4 antibiotics-14-00948-f004:**
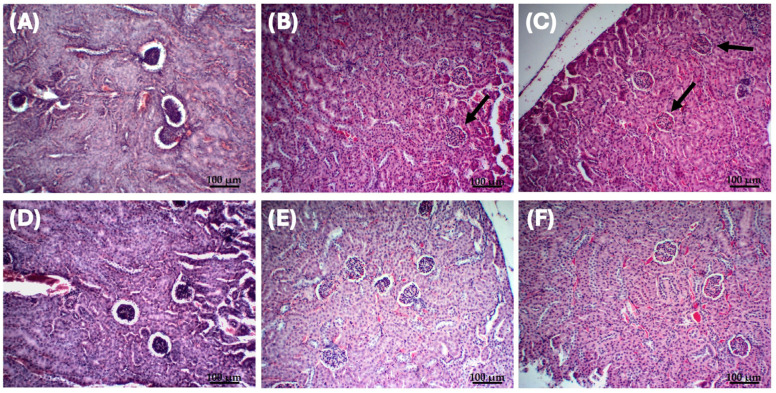
Nephrotoxicity analysis in the Bowman’s space area on day one and after 10 days post-treatment. Black arrows indicate that Bowman’s space areas in the treated groups were contracted on day one. (**A**) Negative control on day one, (**B**) repeated doses toxicity group on day one, (**C**) acute toxicity group on day one, (**D**) negative control after 10 days, (**E**) acute toxicity group after 10 days, and (**F**) repeated doses toxicity group after 10 days. Color: HE. Scale bar: 100 µm.

**Figure 5 antibiotics-14-00948-f005:**
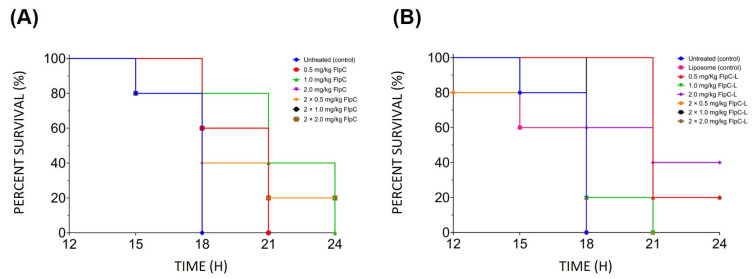
Kaplan–Meier survival curve in mouse bacteremia model caused by *K. pneumoniae* KPN-19 treated intravenously with different doses of (**A**) fluopsin C (Flp) and (**B**) liposomal fluopsin C (Flp-L) DSPE-PEG DMSO+Flp formulation.

**Figure 6 antibiotics-14-00948-f006:**
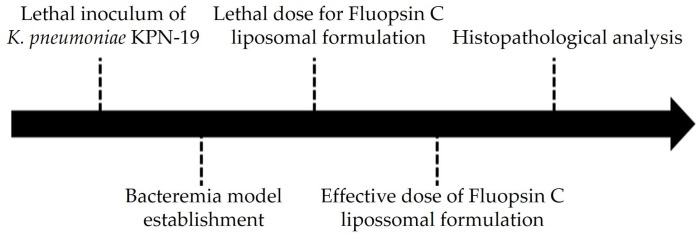
Timeline of the experiment carried out.

**Table 1 antibiotics-14-00948-t001:** Encapsulation efficiency (EE, %) for the different fluopsin C-containing liposomal formulations.

Formulations	EE (%)
SPC+Flp	66.04 ± 8.19
SPC DMSO+Flp	83.75 ± 1.95
DSPE-PEG+Flp	78.04 ± 3.58
DSPE-PEG DMSO+Flp	83.67 ± 2.06

**Table 2 antibiotics-14-00948-t002:** In vitro cytotoxic concentrations of free and liposomal fluopsin C for 50% (CC_50_) and 90% (CC_90_) of the LLC-MK2 cell population by MTT test after 24 h of incubation.

Formulation	CC_50_ (µg/mL)	CC_90_ (µg/mL)
DSPE-PEG DMSO+Flp	1.14	1.26
DSPE-PEG DMSO	>32.00	>32.00
SPC DMSO+Flp	1.26	1.60
SPC DMSO	22.62	79.07
Fluopsin C	0.74	0.85

**Table 3 antibiotics-14-00948-t003:** Composition of fluopsin C-containing liposomal formulations.

Formulation	SPC ^1^ (mM)	Cholesterol (mM)	DSPE-PEG ^2^ (mM)	Fluopsin C (mM)	Phosphate Buffer: DMSO ^3^ (%)
DSPE-PEG+Flp	0.54	0.41	0.05	1	100:0
DSPE-PEG DMSO+Flp	0.54	0.41	0.05	1	90:10
SPC+Flp	0.7	0.3	-	1	100:0
SPC DMSO+Flp	0.7	0.3	-	1	90:10

^1^ Soy phosphatidylcholine (SPC); ^2^ polyethylene glycol 2000 (PEG) conjugated to distearoyl phosphatidyl ethanolamine (DSPE); ^3^ dimethylsulfoxide (DMSO).

## Data Availability

The original contributions presented in the study are included in the article; further inquiries can be directed to the corresponding author.

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
