# Peer review of "Liposomal Fluopsin C: Physicochemical Properties, Cytotoxicity, and Antibacterial Activity In Vitro and over In Vivo MDR Klebsiella pneumoniae Bacteremia Model"

_antibiotics, 2025, doi:10.3390/antibiotics14090948_

Round 1

Reviewer 1 Report

Comments and Suggestions for Authors

My comments are as follow: 

Line 183: Authors should mention the year of the CSLI protocol used in the text.

Line 204: Italicize K. pneumoniae.

Section 2.8.2, 2.8.3, and 2.8.4: Authors should explain why there are differences in the number of mice used.  I suggest that authors add the power calculation for the choice of the number of mice in each mouse experiment.

Line 210: Authors should mention the dosage and composition of the anesthesia used.

In methods, authors should provide a timeline of the animal experiments.

Author Response

1) Line 183: Authors should mention the year of the CSLI protocol used in the text. 

Response: We agree and added the year of the CSLI protocol used (Line 392).

2) Line 204: Italicize K. pneumoniae.

Response: We agree and addressed the error, the style was revised.

3) Section 2.8.2, 2.8.3, and 2.8.4: Authors should explain why there are differences in the number of mice used.  I suggest that authors add the power calculation for the choice of the number of mice in each mouse experiment.

Response: Thank you for the comments. The number of animals used in each group was based on the dichotomous endpoint (survival versus death) for determining the lethal inoculum (4.8.2) and the antibacterial activity of liposomal and free Fluopsin C in vivo (4.8.4). An odd number of animals were used in each analysis, respecting the smallest possible number that still allows a trend to be identified in relation to the outcome, and ethics in the use of animals (reduction of the 3Rs). Except for item 4.8.3, where still based on the principle of reducing the 3Rs, we used the same experimental design as Navarro et al. (n=6) who assessed the lethal dose of free Fluopsin C in female Swiss mice, to compare the results obtained in the present study with the liposomal formulation. We have therefore added the following sentences to the manuscript:

“In a previous study conducted by our research group, under similar experimental conditions [10], the lethal dose (LDâ‚…â‚€) of free Fluopsin C was estimated at 4 mg/kg. These data indicate that the liposomal formulation reduced the toxicity of the compound in mice”. (lines 148-151)

“In a previous study, we evaluated the lethal dose of free Fluopsin C in female Swiss mice [10]. To compare the effect of the liposomal formulation, we used similar experimental conditions. Therefore, the animals were inoculated intravenously with single doses of 0.5, 1, 2, 4, and 8 mg/kg DSPE-PEG DMSO+Flp formulation.” (lines 438-441)

4) Line 210: Authors should mention the dosage and composition of the anesthesia used.

Response: We agree and added specified (Line 429).

5) In methods, authors should provide a timeline of the animal experiments.

Response: Thank you for the comments. We added a figure (6) showing the sequence of animal experiments.

Reviewer 2 Report

Comments and Suggestions for Authors

In this study authors have performed a important work to overcome Antimicrobial resistance. They encapsulated Fluopsin C (Flp) a Metalloantibiotics, in liposomes to enhance efficacy and reduce cytotoxicity. Formulations using soy phosphatidylcholine (SPC) or DSPE-PEG with cholesterol were developed and evaluated for size, stability, drug release, cytotoxicity, and antimicrobial activity against Klebsiella pneumoniae. The DSPE-PEG DMSO+Flp formulation showed superior stability and reduced in vitro cytotoxicity (CCâ‚…â‚€ decreased by 54%). In mice, it effectively reduced mortality from bacteremia by 40% at lower doses, with minimal liver inflammation and mild kidney effects, highlighting its potential against multidrug-resistant K. pneumoniae infections.

Overall the I found the manuscript is well however it is recommended to include a comparative data with other encapsulated Metallic Antibiotics.

Author Response

Reviewer #2 : I would like to sign my review report

1) Overall the I found the manuscript is well however it is recommended to include a comparative data with other encapsulated Metallic Antibiotics.

Response: We do not agree, the comparison between Fluopsin C with other metallic antibiotics does not mean anything. The action is completely different, and it is hard to compare the results. Also, we did not find any articles about encapsulated metallic antibiotics.

Reviewer 3 Report

Comments and Suggestions for Authors

The manuscript in reference aims to evaluate the physicochemical characteristics, cytotoxicity, and antibacterial activity of liposome-encapsulated Fluopsin C (Flp), a natural compound with known antimicrobial properties, against multidrug-resistant (MDR) Klebsiella pneumoniae. The authors claim that the encapsulated Flp exhibits improved stability, reduced toxicity, and maintains its antimicrobial efficacy, making this topic relevant and timely, as it addresses the urgent need for effective agents against MDR pathogens. However, the manuscript has several critical weaknesses that significantly impact the study's clarity, reproducibility, and scientific rigor. In addition, the encapsulation strategy is promising; however, the presentation suffers from a lack of methodological detail, limited statistical depth, overinterpretation, and insufficient novelty.

  1. The novelty of this work is modest. Fluopsin C has been previously reported with liposomal delivery. Here, the formulations are only marginally different from existing work, and the SAR or mechanism of action is not investigated.
  2. The introduction is written in a laconic style and could be improved in composition.
  3. Lines 27–28: Clarify if stability tests followed ICH guidelines. Define what “shelf-life” means in this context.
  4. 75–84: Include passage number, viability check, and storage history of bacterial strains.
  5. Lines 86–90: Purity of Fluopsin C is reported as ">80%" without any chromatogram or clear purity assessment. A compound with only 80% purity introduces uncertainty into all downstream evaluations. This is a critical flaw.
  6. Lines 105–109: Were DMSO-containing formulations tested for leakage or Flp precipitation?
  7. Line 128: “All measurements performed in triplicate” – expand on statistical treatment and batch variability. Indeed, statistical methods (line 258) are shallow. There is no mention of post-hoc corrections, no justification for sample size (power analysis?), and no clear presentation of variance in MIC/MBC or in vivo outcomes.
  8. Line 198–203: Cage assignment and animal identification strategies are verbose but lack core info on randomization or blinding protocols. This is a critical flaw in the design of in vivo studies.
  9. Lines 229–233: Explain why 2 × 2 mg/kg was the only control dose chosen for liposomal formulation.
  10. Figure 5: Mortality curves lack confidence intervals or statistical testing to determine significance between doses.
  11. Table 3: CC50 and CC90 values are given without a confidence interval. Cytotoxicity differences (line 326) are reported as “% reduction” without statistical tests to validate.
  12. Reframe overstatements about efficacy and safety throughout the manuscript.
  13. Lines 332–356: Histopathology reports show mild to moderate necrosis across all treatments, including controls, yet authors interpret this as minimal toxicity. This downplays potential safety concerns.
  14. Lines 370–374: MIC and MBC values between free and liposomal Flp are identical, which contradicts the claim of improved activity. No time-kill curves or synergy tests are provided.
  15. Lines 439–448: The Conclusion is overly enthusiastic. This section should reflect that findings are preliminary and not yet preclinical-grade, and, therefore, the authors must tone down speculative claims in the conclusion.

Comments on the Quality of English Language

Grammar and clarity need improvement. Multiple sentences are overly long or unclear. In addition, several grammatical and stylistic issues are found throughout the manuscript. For instance, “promoting the incorporation of Fluopsin C” → “facilitating Fluopsin C incorporation” or  “Flp properties to develop a new antimicrobial targeting…” → “Flp properties for developing a novel antimicrobial targeting…”

Author Response

  • The novelty of this work is modest. Fluopsin C has been previously reported with liposomal delivery. Here, the formulations are only marginally different from existing work, and the SAR or mechanism of action is not investigated.

Response: Thank you for the comments. But we do not agree. We didn't find any articles on liposomal Fluopsin C in the literature (PubMed, Google Academic). Data on this formulation has been generated by our research group and partially presented at scientific events, such as those cited below.

PEGA, Guilherme E. Almeida et al. Antibacterial activity of Fluopsin C incorporated in different liposomal formulations against multidrug-resistant Staphylococcus aureus. 2021.

DEALIS, Mickely et al. Antibacterial activity of free and liposomal Fluopsin C against multidrug-resistant Staphylococcus aureus BEC 9393. 2021.

  • The introduction is written in a laconic style and could be improved in composition.

Response: We agree and changed it.

  • Lines 27–28: Clarify if stability tests followed ICH guidelines. Define what “shelf-life” means in this context.

Response: About the question of the Reviewer 2, we are just checking how the antibiotic activity of Fluopsin C work. The ICH - International Council for Harmonisation of Technical Requirements for Pharmaceuticals for Human Use, in this time is not usual, and we did not use it.

  • 75–84: Include passage number, viability check, and storage history of bacterial strains.

Response: We agree and included passage number.

  • Lines 86–90: Purity of Fluopsin C is reported as ">80%" without any chromatogram or clear purity assessment. A compound with only 80% purity introduces uncertainty into all downstream evaluations. This is a critical flaw.

Response: Fluopsin C is extracted from F4A fraction which contain four molecules, phenazine hydroxyamide, phenazine hydroxylic, indolinone and Fluopsin C. If Fluopsin C show a presence of contaminant probably is one of these molecules, that does not show any antibiotic activity against Gram-negative bacteria. In this way I do not agree with the Reviewer that suggests a purity of 80% or more introduce uncertainty downstream evaluation. Before the purification process, we checked the purity level by HPLC. Below we show the chromatograms of the Fluopsin C used in those experiments and they were added as Appendix Figure A.3.

  • Lines 105–109: Were DMSO-containing formulations tested for leakage or Flp precipitation?

Response: Fluopsin C shows a high solubility in DMSO.

  • Line 128: “All measurements performed in triplicate” – expand on statistical treatment and batch variability. Indeed, statistical methods (line 258) are shallow. There is no mention of post-hoc corrections, no justification for sample size (power analysis?), and no clear presentation of variance in MIC/MBC or in vivo outcomes.

Response: Thank you for your comments. We have added the following statistical analysis information for the CC50 and CC90 data as Figure A.1 in the Appendix. We have clarified that no variance was found in the MIC/MBC tests, performed in triplicate (we have added this information in lines 192-193). We have also performed a new analysis of the survival data using the Kaplan-Meier estimator for in vivo survival experiments (Figure 5).

  • Line 198–203: Cage assignment and animal identification strategies are verbose but lack core info on randomization or blinding protocols. This is a critical flaw in the design of in vivo studies.

Response: Thank you for your comments and we apologize for the misunderstanding. In fact, the animal experiments were conducted using randomized and blind protocols. We added the following information:

“The animals were randomly allocated to the different experimental groups using a simple randomization protocol. The researchers responsible for administering the treatments/inoculum and assessing the clinical outcomes (including signs of infection and mortality) were blinded as to the identity of the groups. After being assigned to the groups, the animals were kept for a period of 3 days, for the adaptation in the new environment, before each experiment.” (lines 411-416)

  • Lines 229–233: Explain why 2 × 2 mg/kg was the only control dose chosen for liposomal formulation.

Response: 2 × 2 mg/kg is considered the worst-case analysis in this scenario, since in previous tests there was no significant toxicity or antimicrobial activity, it was decided to reduce the number of animals in compliance with ethical issues.

  • Figure 5: Mortality curves lack confidence intervals or statistical testing to determine significance between doses.

Response: Thank you for your comments. We performed a new analysis of the survival data using the Kaplan–Meier estimator. The information has been updated in Figure 5 and line 486.

  • Table 3: CC50and CC90 values are given without a confidence interval. Cytotoxicity differences (line 326) are reported as “% reduction” without statistical tests to validate.

Response: Thank you for your comments and we apologize for the misunderstanding. We added the following information as Appendix Figure A.1.

  • Reframe overstatements about efficacy and safety throughout the manuscript.

Response: Thank you for your feedback regarding the need to re-evaluate the statements about efficacy and safety throughout the manuscript. We fully acknowledge the importance of using precise and balanced language in presenting our results.

Regarding safety: We recognize that the histopathological reports indicated the presence of mild to moderate necrosis in the liver and kidney. We will revise the manuscript to reflect this observation more accurately. It is important to emphasize, however, that mild to moderate necrosis and inflammatory infiltrates in the liver and kidney were also observed in the control groups (negative and liposomal controls). This suggests that necrosis is not exclusively induced by the liposomal formulation.

Our interpretation of an enhanced safety profile—or "minimal toxicity"—is primarily based on the substantial reduction in the toxicity of Fluopsin C when encapsulated in liposomes, compared to the free compound. The DSPE-PEG DMSO+Flp formulation reduced cytotoxicity by 54% and 48% in CC50 and CC90 values, respectively, in vitro. More significantly, no mortality was observed in mice treated with the liposomal formulation at an 8 mg/kg dose, whereas free Fluopsin C exhibited an LD50 of 4 mg/kg. This indicates a safety margin at least twice as high for the encapsulated form.

Furthermore, our analyses demonstrated evidence of organ recovery over time. For the liver, "complete recovery of the organ was observed 40 days after administration of liposomal Fluopsin C." For the kidneys, "after time zero, no significant differences were observed, indicating the recovery of the organ to normal metrics." Therefore, although the presence of necrosis is noteworthy, it appeared to be transient or attenuated when compared to the toxicity profile of free Fluopsin C.

Regarding efficacy: We confirm that encapsulation did not alter the in vitro antimicrobial activity of Fluopsin C; the liposomal formulation retained its ability to combat K. pneumoniae. In vivo, the DSPE-PEG DMSO+Flp formulation demonstrated clear efficacy, reducing mortality by 40% in the mouse bacteremia model. This represents a significant improvement in the survival of infected animals.

We will carefully revise the manuscript to ensure that all statements regarding safety and efficacy are appropriately qualified, emphasizing the relative advantages of liposomal encapsulation (improved safety profile and maintained efficacy), rather than making absolute claims of zero toxicity or complete absence of histopathological damage. Such absolute statements would not be fully supported by the observed mild to moderate necrosis in the control groups. We will focus on a more accurate description of the histopathological findings while maintaining emphasis on the overall therapeutic improvement achieved with Fluopsin C encapsulation.

  • Lines 332–356: Histopathology reports show mild to moderate necrosis across all treatments, including controls, yet authors interpret this as minimal toxicity. This downplays potential safety concerns.

Response: Thank you for your pertinent observation regarding the histopathological findings of necrosis. We would like to justify our interpretation of “minimal toxicity” based on the following points:

  • Presence of Necrosis in Control Groups: Histopathological analysis of both the liver and kidney revealed mild to moderate necrosis across all groups, including the negative and liposomal controls. Appendix Table A1 confirms that both the "Liposomal Control" and "Negative Control" groups exhibited necrosis scores, indicating that this finding is not exclusive to the encapsulated Fluopsin C treatment.

  • Organ Recovery Over Time: Despite the initial presence of necrosis, the study demonstrated organ recovery. For the liver, "complete recovery of the organ was observed 40 days after administration of liposomal Fluopsin C," and the increase in binucleated hepatocytes in all treatment groups (except the negative control) "suggests that the liver may recover following antimicrobial treatment." Regarding the kidneys, after the first day of treatment, "no significant differences were observed, indicating recovery of the organ to normal metrics.”

  • Significant Reduction in Toxicity Compared to Free Fluopsin C: The primary objective of liposomal encapsulation was to "reduce the compound cytotoxicity" of Fluopsin C. In in vitro tests, the DSPE-PEG DMSO+Flp formulation reduced cytotoxicity by 54% and 48% in CC50 and CC90 values, respectively, compared to free Fluopsin C. Notably, in vivo, "no mortality was associated with the use of this formulation at 8 mg/kg," whereas free Fluopsin C presented an LD50 of 4 mg/kg in mice. This indicates that the lethal dose of the liposomal formulation is at least double that of the free compound. Furthermore, the nephrotoxicity profile of the liposomal formulation "differed from that of free Fluopsin C," presenting necrosis but lacking other types of lesions observed in the free compound group.

Therefore, our interpretation of “minimal toxicity” is based not only on the isolated finding of necrosis, but primarily on the substantial improvement in the safety profile compared to free Fluopsin C, and on the demonstrated capacity for organ recovery throughout the study.

  • Lines 370–374: MIC and MBC values between free and liposomal Flp are identical, which contradicts the claim of improved activity. No time-kill curves or synergy tests are provided.

Response: We did not evaluate a synergy between Fluopsin C and other molecule, also we tested the lipossome alone and did not show any activity or influence on Fluopsin C activity.

  • Lines 439–448: The Conclusion is overly enthusiastic. This section should reflect that findings are preliminary and not yet preclinical-grade, and, therefore, the authors must tone down speculative claims in the conclusion.

Response: We agree and changed.

Reviewer 4 Report

Comments and Suggestions for Authors

The original manuscript, entitled “Liposomal Fluopsin C: physicochemical properties, cytotoxicity, and antibacterial activity in vitro and over in vivo MDR Klebsiella pneumoniae bacteremia model” describes the liposomal formulations of fluopsin C, a broad-spectrum antibacterial agent, and their physiochemical and biological evaluations. The DSPE-PEG DMSO+Flp formulation presented superior physicochemical stability and maintained antimicrobial activity. Low and moderate toxicity of this formulation was observed, suggesting its potential against infections by MDR K. pneumoniae. However, some points require revision as outlined below.

  1. More information about physiochemical property of fluopsin C such as structure, solubility, MIC and MBC in the Introduction section.
  2. Specify the amount of liposome or fluopsin C used in the characterization such as ZP and drug release.
  3. Give the concentration range of fluopsin C used in calibration curve (line 136)
  4. It is better to provide the results of analytical method validation such as accuracy, precision, LOD, and LOQ for method used in drug release.
  5. Give the details of the original of the LLC-MK2 cell (line 162).
  6. Please clarify the concentrations (32 – 0.25 ug/mL). Is it the concentration of liposome or that equivalent to fluopsin C?
  7. The authors should discuss the relationship between zeta potential and diameter after doing the stability study.
  8. The result of the drug release was reported until 11 h of the experiment. The authors should give more information in text until the end of the study.
  9. Give the SD of data in Table 3.
  10. Why did the only DSPE-PEG DMSO+Flp use in vivo toxicity and bacteremia assay. It seems that SPC DMSO+Flp had higher CC50 or drug release. Although the authors stated that DSPE-PEG DMSO+Flp was identified as the most promising formulation due to the advantages over the immune system, it is better to provide the strong evidence supporting this study.
  11. Please provide the limitations of the study.

Author Response

  1. More information about physiochemical property of fluopsin C such as structure, solubility, MIC and MBC in the Introduction section.

Response: We agree and was included more details about FlpC knowledge in the Introduction section (Lines 50-58)

2) Specify the amount of liposome or fluopsin C used in the characterization such as ZP and drug release.

Response: We agree and adding specification (Lines 323).

3) Give the concentration range of fluopsin C used in calibration curve (line 136)

Response: We agree and adding specification (Line 338-339).

4) It is better to provide the results of analytical method validation such as accuracy, precision, LOD, and LOQ for method used in drug release.

Response: We agree and add more clarified information considering that the method used represents the standard model (Line 337).

5) Give the details of the original of the LLC-MK2 cell (line 162).

Response: We agree and adding specification (Line 369).

6) Please clarify the concentrations (32 – 0.25 ug/mL). Is it the concentration of liposome or that equivalent to fluopsin C?

Response: We agree and added more clarified information (Line 376).

7) The authors should discuss the relationship between zeta potential and diameter after doing the stability study.

Response: We agree and have added more clarified information in the discussion section (Lines 226-231).

8) The result of the drug release was reported until 11 h of the experiment. The authors should give more information in text until the end of the study.

Response: We agree and we included more details about FlpC release (Lines 127-130).

9) Give the SD of data in Table 3.

Response: Thank you for your comments and we apologize for the misunderstanding. We added more information about de statistical results, the following information as Appendix Figure A.1.

10) Why did the only DSPE-PEG DMSO+Flp use in vivo toxicity and bacteremia assay. It seems that SPC DMSO+Flp had higher CC50 or drug release. Although the authors stated that DSPE-PEG DMSO+Flp was identified as the most promising formulation due to the advantages over the immune system, it is better to provide the strong evidence supporting this study.

Response: Thank you for your comments, this information about the criteria used to choose the best liposomal formulation is consolidated knowledge in the bibliography and is included in the Discussion section.

"In in vivo system, higher difference of release was expected among liposomal formulations, due to the presence of plasmatic proteins, increasing the potential of the DSPE-PEG DMSO+Flp formulation. In the absence of PEG of liposomal formulation, plasmatic proteins opsonize the liposomes, leading to clearance by the mononuclear phagocytic system [27, 29]. Previous studies on antimicrobial liposome encapsulation have found similar results like us, including encapsulation efficiency [30-33]." (Lines 239-244).

"However, by other authors, this formulation could lead to fast opsonization, phago-cytosis, and removal from circulation by the action of the mononuclear phagocyte system, presenting a potential risk for this formulation [29]. DSPE-PEG DMSO+Flp was identified as the most promising formulation due to the advantages over the immune system, [...]" (Lines 246-250).

11) Please provide the limitations of the study.

Response: We are not sure which kind of limitations you are seeing. The results were repeated many times with similar results. The use of liposomes decreased cytotoxicity and kept the antibiotic activity. The limitation is the production of Fluopsin, the LV strain produce small amount of the compound, and we need to use organic solvent in large amounts. The bacteria control is obviously and soon certainly the Fluopsin should be used to help control MDR strains.

Round 2

Reviewer 3 Report

Comments and Suggestions for Authors

The authors have adequately addressed my comments and successfully clarified the concerns raised during peer review; therefore, the manuscript can now proceed in the editorial process.

Comments on the Quality of English Language

The manuscript requires a detailed scrutiny to revise some grammar and stylistic issues throughout the document.